# CADSim: Robust and Scalable in-the-wild 3D Reconstruction for Controllable Sensor Simulation

**Jingkang Wang**[1,2]   **Sivabalan Manivasagam**[1,2]   **Yun Chen**[1,2]   **Ze Yang**[1,2]
**Ioan Andrei Bârsan**[1,2]   **Anqi Joyce Yang**[1,2]   **Wei-Chiu Ma**[1,3]   **Raquel Urtasun**[1,2]

Waabi[1]   University of Toronto[2]   Massachusetts Institute of Technology[3]

{wangjk,manivasagam,zeyang,yun,iab,ajyang,urtasun}@cs.toronto.edu   weichium@mit.edu

**Abstract:** Realistic simulation is key to enabling safe and scalable development of self-driving vehicles. A core component is simulating the sensors so that the entire autonomy system can be tested in simulation. Sensor simulation involves modeling traffic participants, such as vehicles, with high quality appearance and articulated geometry, and rendering them in real time. The self-driving industry has typically employed artists to build these assets. However, this is expensive, slow, and may not reflect reality. Instead, reconstructing assets automatically from sensor data collected in the wild would provide a better path to generating a diverse and large set with good real-world coverage. Nevertheless, current reconstruction approaches struggle on in-the-wild sensor data, due to its sparsity and noise. To tackle these issues, we present *CADSim*, which combines part-aware object-class priors via a small set of CAD models with differentiable rendering to automatically reconstruct vehicle geometry, including articulated wheels, with high-quality appearance. Our experiments show our method recovers more accurate shapes from sparse data compared to existing approaches. Importantly, it also trains and renders efficiently. We demonstrate our reconstructed vehicles in several applications, including accurate testing of autonomy perception systems.

**Keywords:** 3D Reconstruction, CAD models, Sensor Simulation, Self-Driving

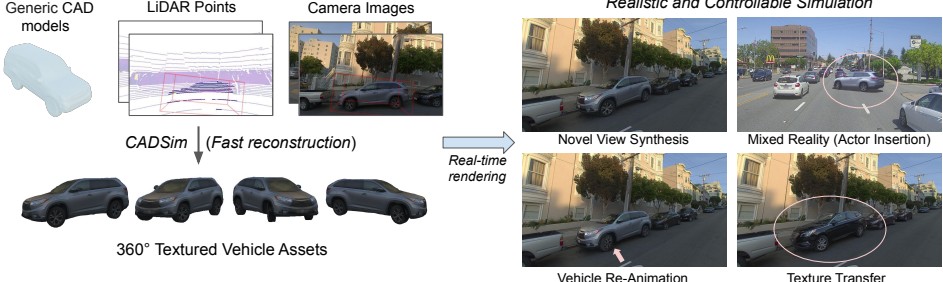

Figure 1: **CADSim recovers shape, appearance and illumination in a robust and scalable way from sensor observations**. The reconstructed assets are of high fidelity, part-aware, geometry-aligned and compatible with graphics engine, enabling efficient, realistic and controllable simulation.

## 1 Introduction

Robots, such as self-driving vehicles (SDVs), learn to navigate environments and interact safely with other agents through experience. It is critical for robots to handle rare or safety-critical situations which are challenging or dangerous to observe in the real world [1]. Simulation provides a solution for the robot to efficiently experience edge-case scenarios for evaluation in a safe and cost-effective manner. For proper testing that covers the full space of possible scenarios, the simulator should have a diverse and large set of traffic participants. For example, vehicles have a wide variety of shapes, sizes, and appearances. We want to ensure that the perception and autonomy systems can detect them and act appropriately in all possible situations. Existing self-driving simulators have a limited set of objects, as each 3D asset is typically designed manually, with fully specified shape, appearance, etc. This is a time-consuming and expensive process that does not scale. In this work, we propose to leverage real sensor data collected by the SDV in the wild to automatically build an asset library

6th Conference on Robot Learning (CoRL 2022), Auckland, New Zealand.

for realistic sensor simulation. In-the-wild data is cost-efficient and scalable to obtain, enabling the collection of a large and diverse set of objects.

Specifically, given a sequence of sensor data (camera images and LiDAR point clouds) collected by an SDV, our goal is to automatically create digital replicas for all nearby vehicles in the scene and directly use them in a high-fidelity simulator to create many scenarios that never existed before. To achieve high-quality and scalable sensor simulation, the reconstructed assets should (i) have precise shape and photorealistic appearance; (ii) be easily editable to create new variations from the original assets, (iii) be controllable, such as the ability to manipulate the vehicle to generate new behaviors (*e.g.*, wheels steering) and (iv) allow for real-time rendering, which is important for closed-loop training and large-scale evaluation [2]. Out of past works using various 3D representations for reconstruction, such as point clouds [3, 4, 5, 6], voxels [7, 8, 9], or neural implicit representations [10, 11, 12, 13], the mesh representation best achieves the desired properties [14, 15]. In this paper we utilize mesh representations since they are widely used throughout simulation [16, 17, 18] and content creation systems [19, 20], are easy to manipulate [21, 22], suitable for rigging and animation [23, 24, 25, 26], support texturing and material properties [27, 28], and are fast to render [29].

Generating meshes of nearby objects from in-the-wild sensor data is a challenging task. The sensor data is sparse ($< 30$ views) and may be noisy. Furthermore, the objects of interest are observed from limited viewpoints and portions of the objects are unobserved, whereas we want to reconstruct the complete shape for simulation. Previous mesh reconstruction methods tackle this task as an energy minimization problem, where given an initial sphere [30] or "mean-shape" mesh template [31, 32], the goal is to optimize the mesh to be consistent with sensor data, subject to some regularization. However, when operating on in-the-wild sensor data, these methods generate overly smoothed meshes that are inaccurate [30, 32], or have severe artifacts and self-intersecting edges [33]. We demonstrate in our experiments that this is due to poor initialization. Moreover, these approaches generate rigid meshes that cannot be controlled and lack articulation. We therefore propose a simple and effective solution—to leverage class-specific CAD models to provide priors for reconstruction. Not only do CAD models have high quality shape, they encode rich semantic information, such as where the wheels in a vehicle are and how they move. This enables better initialization and reconstruction of high-quality *articulated* meshes from real-world data.

CAD models alone are not sufficient for realistic sensor simulation, as we we want to create digital twins of any possible vehicle. Instead, we optimize our vehicle CAD model representation with a state-of-the-art differentiable renderer in an energy minimization framework. This allows us to faithfully reconstruct a diverse set of real-world vehicles, including their moving parts (*e.g.*, wheels), from very sparse and noisy observations. We compare against a wide range of reconstruction methods, including recent neural rendering-based approaches, and show improved reconstruction performance on both camera and LiDAR sensor data. We show that our reconstructed vehicles can be used for a variety of applications such as novel view synthesis, texture transfer, mixed reality via injecting simulated actors into real data, and vehicle re-animation (Fig. 1). We also demonstrate that our approach creates a more realistic simulation environment for robots such as SDVs by evaluating perception models on simulated sensor data and showing small domain gap compared to real data.

## 2 Related Work

**3D Representations for Sensor Simulation.** Triangular meshes are currently the de facto standard in real-time graphics [14, 15, 16]. While fast, flexible, compact, and widely adopted in game engines and simulators [16], mesh-based assets are usually regarded as expensive to create due to the need for human artists. Surfels [34] have also been leveraged in LiDAR simulation for robotic applications [35, 36]. However, their lack of geometric structure makes them difficult to animate, edit, and compress. Recently, volumetric representations have experienced a surge in popularity [37, 38, 39] thanks to their simplicity, differentiability, and flexibility in modeling non-Lambertian effects [40, 41, 42]. These works can roughly be divided into neural volumetric rendering, such as NeRFs [40, 43, 44, 45, 46], and implicit surfaces such as signed distance fields [47, 48, 49] or occupancy fields [10, 50, 51]. Various methods have been proposed to speed up surface and volumetric rendering [52, 53, 54, 55, 56]. However, these approaches still lag behind traditional rasterization pipeline in terms of rendering speed and throughput. In our work, we use triangular meshes optimized from CAD models for efficient and accurate geometry modelling.

**Multi-View 3D Reconstruction in the Wild.** Despite the incredible success achieved by recent neural volumetric representations, these works are mainly demonstrated on synthetic datasets with

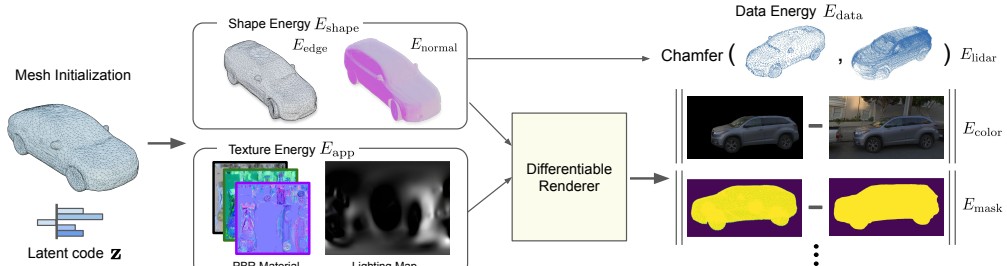

Figure 2: Overview of CADSim for 3D Reconstruction on in-the-wild data.

hundreds of (noise-free) images and dense coverage on the camera viewpoints [40, 57, 30]. However, the real world input is usually much sparser (*e.g.*, few views of a single object with similar view points) and noisier (*e.g.*, localization, calibration error and LiDAR noise) for real world applications such as robotics and self-driving. To address this issue, recent works [58, 59, 60] proposed conditional models that require expensive pre-training on other scenes with multi-view images and camera pose annotations. Another recent line of work proposes learning mesh representations directly, either by deforming a generic mesh [61, 27, 62, 33, 63, 64], or by leveraging a differentiable mesh extraction algorithm on top of an implicit representation [65, 66, 67]. Different from past works, we show that simple mesh optimization with differentiable rendering is sufficient for high-fidelity reconstruction.

**CAD Priors.** Retrieving a 3D model conditioned on a single observation has been shown to be a strong baseline which outperforms more sophisticated early approaches to single-view reconstruction [68]. This motivates the use of imperfect but readily available 3D CAD models as a powerful prior for reconstruction [69, 31, 70, 71, 72, 73, 74, 75, 76]. [69, 31, 71] use the truncated SDF volumes to build the latent space for the CAD models for joint pose and shape estimation. Uy et al. [74, 75] encode sparse observations and use them to retrieve the most similar candidate from a CAD asset bank, which is then deformed to match the input. While promising, none of the approaches in this area have been demonstrated to work outside of synthetic [77] datasets. Most do not model realistic texture, nor are they part-aware. Lu et al. [76] conduct a two-stage CAD-based deformation and handle the wheels separately. However, they recover the appearance only from single images without physics based materials and lighting models, and do not optimize the articulation from data.

## 3 Method

Given camera images and/or LiDAR point clouds collected by a self-driving car, our goal is to automatically create digital replicas for all nearby vehicles in the scene and use them in a high-fidelity simulator. We build our model based on the observation that existing CAD models are equipped with detailed geometry and animatable parts, which can be used as a prior during the reconstruction process. Towards this goal, we propose an energy-based formulation that exploits CAD models, as well as visual and geometric cues from images and/or LiDAR for 3D reconstruction (Fig. 2). We first describe our model representation and how it leverages CAD priors (Sec. 3.1). Then we present our energy-based model (Sec. 3.2), and how we conduct inference to generate the vehicle mesh (Sec. 3.3).

### 3.1 Model Representation

To faithfully simulate an object, we need to recover not only its geometry, but also its material. Towards this goal, we exploit *textured meshes* as underlying representation since they best meet the desired properties of modern high-fidelity simulators, by being efficient to animate, edit, and render.

**Mesh representation.** A mesh $\mathcal{M} = (\mathbf{V}, \mathbf{F})$ is composed of a set of vertices $\mathbf{V} \in \mathbb{R}^{|V| \times 3}$ and faces $\mathbf{F} \in \mathbb{N}^{|F| \times 3}$, where the faces define the connectivity of the vertices. Our goal is to deform a mesh to match the observations from the sensor data. Typically, during deformation, the topology (*i.e.*, connectivity) of the mesh is fixed and only the vertices are "moving" [23, 32, 78, 21]. This strategy greatly simplifies the deformation process, yet at the same time constrains its representation power. For instance, if the original mesh topology is relatively simple (*e.g.*, a sphere [78] or a "mean-shape" mesh template [32]) and non-homeomorphic to the object of interest, the mesh may struggle to capture the fine-grained geometry (*e.g.*, side mirrors of a car).

**CAD models as priors.** To circumvent such limitations, we propose to incorporate shape priors from CAD models into the mesh reconstruction. One straightforward approach is to directly exploit

the CAD model to initialize the mesh $\mathcal{M}_{\text{CAD}} = (\mathbf{V}_{\text{CAD}}, \mathbf{F}_{\text{CAD}})$. Since CAD models, by design, respect the topology of real-world objects, the mesh will be able to model finer details. One obvious caveat, however, is that there is no structure among the vertices. Each vertex may move freely in the 3D space during the optimization; *e.g.*, there is no guarantee that vertices belonging to a wheel will continue to form a cylinder. We thus further incorporate the *part information* from CAD models into the parameterization. This allows us to *inherently* preserve part geometry, as well as enable part manipulation (*e.g.*, changing the steering of the wheels).

We leverage the semantic part information from CAD models to partition the vehicle mesh $\mathcal{M}_{\text{CAD}}$ into a vehicle body $\{\mathbf{V}_{\text{body}}, \mathbf{F}_{\text{body}}\}$ and wheels $\{\mathbf{V}_{\text{wheel}}^{(k)}, \mathbf{F}_{\text{wheel}}^{(k)}\}_{i=1}^{K}$. $K$ indicates the number of wheels a vehicle has. $K = 4$ for most vehicles but may be 6 or more for large trucks. We employ the fact that the wheels are

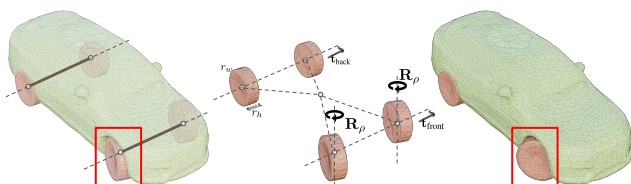

Figure 3: **Vehicle model used in CADSim**. For simplicity, we only show the offsets along the axle axis.

usually the same within the same vehicle, and model all wheels with the same underlying mesh $\{\mathbf{V}_{\text{wheel}}, \mathbf{F}_{\text{wheel}}\}$ and their individual relative pose $\mathbf{T}^k$ to the vehicle origin. As there are a wide variety of vehicles with different wheel sizes and different relative positions to the vehicle body, we further add a scale factor $\mathbf{r} = [r_w, r_h, r_w]$ (wheel radius and thickness), and per-axle translation offsets $\mathbf{t}_{\text{front}}, \mathbf{t}_{\text{back}} \in \mathbb{R}^3$ with respect to the wheel origin. We can thus drastically reduce the degrees of freedom, enforce the shape of the wheel, and guarantee the relative positions of the wheels to be symmetric. Importantly, the front-axle wheels can be steered and do not necessarily align with the body. We hence parameterize the front wheels to have a yaw-relative orientation $\rho$. The full vehicle mesh model (see Figure 3) can thus be written as:

$$\mathbf{V}_{\text{wheel}}^k(\mathbf{r}, \mathbf{t}_{\text{front}}, \rho; \mathbf{V}_{\text{wheel}}) = \mathbf{T}^k(\mathbf{R}_\rho \mathbf{r} \mathbf{V}_{\text{wheel}} + \mathbf{t}_{\text{front}}) \quad k \in \{1, 2\}, \quad \text{(front wheels)}$$

$$\mathbf{V}_{\text{wheel}}^k(\mathbf{r}, \mathbf{t}_{\text{back}}; \mathbf{V}_{\text{wheel}}) = \mathbf{T}^k(\mathbf{r} \mathbf{V}_{\text{wheel}} + \mathbf{t}_{\text{back}}) \quad k \in \{3, 4, ..., K\}, \quad \text{(remaining wheels)} \quad (1)$$

$$\mathbf{V} = \{\mathbf{V}_{\text{body}}, \mathbf{V}_{\text{wheel}}^{(k)}\} \quad k \in \{1, \ldots, K\},$$

with $\mathbf{V}_{\text{body}}, \mathbf{r}, \mathbf{t}_{\text{back}}, \mathbf{t}_{\text{front}}, \rho$ being the free variables.

**Extending to a CAD library.** While a CAD model provides a useful prior for mesh initialization, it is likely that a single template mesh may not sufficiently cover a wide range of objects. We thus resort to a CAD model library that consists of various vehicle types. Inspired by [31, 76], we represent a large variety of vehicles with a compact, low dimensional code, which allows us to quickly determine the best CAD model to initialize from. We first parameterize a collection of the CAD models with Eq. (1), simplify them so that they have the same number of vertices and topology, and align them in a shared coordinate space with dense correspondences. Finally, we apply principal component analysis on vertex coordinates to obtain a shared low-dimensional code $\mathbf{z}$. Note that since all CAD models are parameterized with Eq. (1), the deformed mesh will reflect this part structure as well. Please refer to the supp. material for more details.

**Appearance representation.** In addition to geometry, our mesh representation must accurately capture how light interacts with its surface so that we can realistically reproduce sensor observations. Towards this goal, we parameterize the mesh appearance using a physically-based appearance representation. Specifically, we represent appearance using a micro-facet BRDF model with the differentiable split sum environment lighting [79] proposed by Munkberg et al. [67]. Since the topology of the mesh is fixed, we can build a one-to-one mapping relationship between each point on the mesh surface and each point in the 2D $(u, v)$ space. We use a physically-based (PBR) material model from Disney [80], which contains a diffuse lobe $\mathbf{k}_d$ with an isotropic, specular GGX lobe [81].

### 3.2 Energy Formulation

The next step after the CAD model initialization is to optimize both the geometry and the appearance of the mesh such that it best matches the captured sensor data. Our approach builds upon recent success on differentiable rendering [82, 83, 84, 85], leveraging a differentiable renderer that takes as input variables such as the sensor pose and our textured mesh representation, and outputs a realistic simulation of the object. Differentiable rendering allows all variables to be optimized based on the supervision provided by the input images.

**Notation.** Let $\mathcal{I} = \{\mathbf{I}_i\}_{1 \leq i \leq N}$ be the images captured at different timestamps and $\mathcal{P}$ be the aggregated LiDAR point clouds captured by a data collection platform, in our case a self-driving vehicle driving in the real world. Let $\{\mathbf{M}_i\}_{1 \leq i \leq N}$ be the foreground segmentation mask of $\{\mathbf{I}_i\}_{1 \leq i \leq N}$ obtained from an off-the-shelf algorithm [30, 86]. Let $\mathcal{A} = \{\mathcal{D}, \mathcal{R}\}$ be the variables directly related to the appearance model (*i.e.*, the material and the lighting). We denote $\Pi = \{\mathbf{K}_i^{\text{cam}}, \boldsymbol{\xi}_i^{\text{cam}}, \boldsymbol{\xi}^{\text{lidar}}\}$ as the intrinsics and the extrinsics of the sensors, where $\boldsymbol{\xi} \in \mathfrak{se}(3)$ are elements of the Lie algebra associated with SE(3). We assume all cameras are pre-calibrated with known intrinsics. Let $\psi : (\mathcal{M}, \mathcal{A}, \Pi) \to (\mathbf{I}_\psi, \mathbf{M}_\psi)$ be the differentiable renderer where $\mathbf{I}_\psi$ and $\mathbf{M}_\psi$ denote the rendered RGB image and object mask.

**Overall energy.** We design an energy function with complementary terms which measure the geometry and appearance agreement between the observations and estimations ($E_{\text{data}}$), while regularizing the shape ($E_{\text{shape}}$) and appearance ($E_{\text{app}}$) to obey known priors:

$$\underset{\mathcal{M},\Pi,\mathcal{A}}{\text{argmin}} \{E_{\text{data}}(\mathcal{M}, \Pi, \mathcal{A}; \mathcal{I}, \mathcal{P}) + \lambda_{\text{shape}} E_{\text{shape}}(\mathcal{M}) + \lambda_{\text{app}} E_{\text{app}}(\mathcal{M}, \Pi, \mathcal{A}; \mathcal{I}, \mathcal{P})\}. \quad (2)$$

We now describe each energy term in more detail.

**Data term.** This energy encourages the estimated textured mesh to match the sensor data as much as possible. Specifically, it consists of three components:

$$E_{\text{data}}(\mathcal{M}, \Pi, \mathcal{A}; \mathcal{I}, \mathcal{P}) = E_{\text{color}}(\mathcal{M}, \Pi, \mathcal{A}; \mathcal{I}) + \lambda_{\text{mask}} E_{\text{mask}}(\mathcal{M}, \Pi; \mathcal{I}) + \lambda_{\text{lidar}} E_{\text{lidar}}(\mathcal{M}, \Pi; \mathcal{P}). \quad (3)$$

$E_{\text{color}}, E_{\text{mask}}$ both enforce the rendering of the textured mesh to be close to the image observations. They, however, are complementary. $E_{\text{color}}$ encourages the appearance of the rendered image to match the RGB observation and propagates the gradients to all variables including appearance variables $\mathcal{A}$, while $E_{\text{mask}}$ measures the mask difference and only depends on the shape of the mesh as well as the camera poses. Following previous work [30, 67], we exploit smooth-$\ell_1$ distance to measure the difference in RGB space and squared $\ell_2$ for object masks:

$$E_{\text{color}} = \frac{1}{N} \sum_i^N \bar{\ell}_1 \left(\mathbf{I}_\psi(\mathcal{M}, \mathbf{K}_i, \boldsymbol{\xi}_i, \mathcal{A}), \mathbf{I}_i\right) \qquad E_{\text{mask}} = \frac{1}{N} \sum_i^N \|\mathbf{M}_\psi(\mathcal{M}, \mathbf{K}_i, \boldsymbol{\xi}_i) - \mathbf{M}_i\|_2^2, \quad (4)$$

with $\bar{\ell}_1$ the smooth-$\ell_1$ norm [87] and $N$ the number of images available.

$E_{\text{lidar}}$ encourages the geometry of our mesh to match the aggregated LiDAR point clouds. Since minimizing point-to-surface distance is computationally expensive in practice, we adopt the popular Chamfer Distance (CD) to measure the similarity instead. We randomly select $L$ points ($\mathcal{P}_s$) from the current mesh and compute the asymmetric CD of $\mathcal{P}_s$ with respect to the aggregated point cloud $\mathcal{P}$:

$$E_{\text{lidar}} = \text{CD}(\mathcal{P}, \mathcal{P}_s) = \frac{1}{|\mathcal{P}|} \sum_{x \in \mathcal{P}} \alpha_x \min_{y \in \mathcal{P}_s} \|x - y\|_2^2, \quad (5)$$

where $\alpha$ is an indicator function representing which LiDAR point is an outlier. We refer the reader to Sec. 3.3 for a discussion on how to estimate $\alpha$.

**Shape term.** This energy encourages the deformed mesh to be smooth and the faces of the mesh to be uniformly distributed among the surfaces (so that the appearance would be less likely distorted). Specifically, $E_{\text{shape}}$ is an addition of two shape regularizations [85, 63, 30]: a normal consistency term $E_{\text{normal}}(\mathbf{V})$ and an average edge length term $E_{\text{edge}}(\mathbf{V})$:

$$E_{\text{normal}}(\mathbf{V}) = \frac{1}{N_{\mathbf{F}}} \sum_{\mathbf{f} \in \mathbf{F}} \sum_{\mathbf{f}' \in \mathcal{N}(\mathbf{f})} \|\mathbf{n}(\mathbf{f}) \cdot \mathbf{n}(\mathbf{f}')\|_2^2 \qquad E_{\text{edge}}(\mathbf{V}) = \frac{1}{N_{\mathbf{E}}} \sum_{\mathbf{v} \in \mathbf{V}} \sum_{\mathbf{v}' \in \mathcal{N}(\mathbf{v})} \|\mathbf{v} - \mathbf{v}'\|_2^2. \quad (6)$$

Here, $\mathbf{f} \in \mathbf{F}$ and $\mathbf{v} \in \mathbf{V}$ refer to a single face and a single vertex. $\mathcal{N}(\mathbf{f})$ and $\mathcal{N}(\mathbf{v})$ denote the neighboring faces of $\mathbf{f}$ and the neighboring vertices of $\mathbf{v}$ respectively. $N_{\mathbf{F}}$ and $N_{\mathbf{E}}$ are the number of neighboring face pairs and edges.

**Appearance term.** This energy exploits the following facts: (1) the appearance of a vehicle will not change abruptly in most cases; instead, it varies in a smooth fashion; (2) neutral, white lighting dominates in the real world. Following prior art in intrinsic decomposition [88, 89], we adopt a sparsity term to penalize frequent color changes on the diffuse $\mathbf{k}_d$ and specular $\mathbf{k}_s$ terms. We also add a regularizer to penalize the environment light in gray scale.

$$E_{\text{app}} = \lambda_{\text{mat}} \left(\|\nabla \mathbf{k}_d\|_1 + \|\nabla \mathbf{k}_s\|_1\right) + \lambda_{\text{light}} \sum_{i=1}^3 \|\mathbf{c}_i - \bar{\mathbf{c}}_i\|_1, \quad (7)$$

where $\nabla \mathbf{k}_d$ and $\nabla \mathbf{k}_s$ are the image gradients approximated by Sobel-Feldman operator [90]. $\mathbf{c}_i$ and $\bar{\mathbf{c}}_i$ are the light intensity values at R,G,B channels and the per-channel average intensities.

## 3.3 Inference

Our goal is to find the optimal mesh $\mathcal{M}^*$, appearance representation $\mathcal{A}^*$, and sensor poses $\Pi^*$ such that the total energy is minimized. We adopt the following strategy for better inference convergence.

**Initialization.** A good initialization is required for non-convex optimization-based methods to achieve good performance. If the energy model is initialized from a CAD asset that is very different from the observations, the mesh may not be able to fit the sensor data well. To handle a wide variety of vehicles, we leverage the learned low-dimensional latent code $\mathbf{z}$ to estimate the initial coarse mesh $\mathcal{M}_{\mathrm{init}} = (\mathbf{V}_{\mathrm{init}}, \mathbf{F})$, where $\mathbf{V}_{\mathrm{init}}$ is reconstructed from the optimized latent code $\mathbf{z}^*$ and

$$z^* = \operatorname*{argmin}_{z} \lambda_{\mathrm{mask}} E_{\mathrm{mask}}(\mathcal{M}, \Pi; \mathcal{I}) + \lambda_{\mathrm{lidar}} E_{\mathrm{lidar}}(\mathcal{M}, \Pi; \mathcal{P}) + \lambda_{\mathrm{shape}} E_{\mathrm{shape}}(\mathbf{V}). \quad (8)$$

We note that we focus on the geometry and only optimize the latent code. The latent code is initialized from $\mathbf{0}$, and the sensor poses are obtained from coarse calibration and fixed. We optimize $\mathbf{z}$ using stochastic gradient descent with the Adam optimizer [91]. Please see the supp. material for additional inference details. Since the low-dimensional latent code already smooths out instance-specific details, the optimization is much more robust and less sensitive to initialization.

**Efficient and robust inference.** Given the initialization $\mathcal{M}_{\mathrm{init}}$, we jointly optimize the vertices $\mathbf{V}$, appearance variables $\mathcal{A}$, and sensor poses $\Pi$ as described in Eq. (2). To save computation, we uniformly sample $L$ points on the current mesh at each iteration to compute the LiDAR energy $E_{\mathrm{lidar}} = \mathrm{CD}(\mathcal{P}, \mathcal{P}_s)$, where $\mathcal{P}_s \sim \mathcal{M}$ and $|\mathcal{P}| = L$. To handle LiDAR outliers, we estimate the indicator function $\alpha$ by only calculating the Chamfer distance for the top $p$ percentage of point pairs with the smallest distance. We find it is sufficient to handle point outliers in most cases. For the sensor extrinsics, we optimize the 6D rotation representation [92] and 3D translation. It is a well-known issue that optimizing vertex positions with adverse gradient descent steps can cause self-intersections that is irrecoverable or even exacerbated in further steps [63, 33]. Therefore, following Nicolet et al. [33], we use preconditioned gradient descent steps to bias towards smooth solutions at each iteration, enabling faster convergence to higher-quality meshes with more fine-grained details.

## 4 Experiments

### 4.1 Experimental Setup

**Datasets.** We evaluate our approach on two *in-the-wild* datasets: Multi-View Marketplace Cars (MVMC) [30] and PandaVehicle [93]. MVMC consists of a diverse set of vehicles captured under various illumination conditions. Each vehicle comes with around 10 views and rough camera pose. Following [30], we evaluate our model as well as the baselines on the selected 20 vehicles. PandaVehicle is a dataset we derived from the PandaSet dataset [93]. It consists of 200+ vehicles, along with their corresponding multi-view images and aggregated LiDAR point clouds. We manually inspect and select 10 high-quality vehicles for evaluation based on the accuracy of LiDAR-camera calibration. On average, each vehicle is captured from 24 camera views.

**Baselines.** We compare our approach with three types of 3D reconstruction methods: (1) *Neural radiance fields*: NeRF++ [43], and Instant-NGP [46]. (2) *Implicit surface representations*: NeRS [30], NVDiffRec [67], and NeuS [48]. (3) *Explicit geometry-based approaches*: single-image view-warping (SI-ViewWarp) [94], which warps the source image to the novel target view using depth estimated by LiDAR points; and multi-image view-warping (MI-ViewWarp), which blends multiple source images to the target view. We also compare against SAMP [31], a CAD model mesh optimization approach which leverages SDF-aligned CAD models for joint pose and shape optimization.

**Evaluation settings and metrics.** We evaluate reconstruction quality by performing camera and LiDAR simulation with the reconstructed vehicle. For camera simulation (*i.e.*, novel view synthesis (NVS)), we adopt mean-square error (MSE), peak signal-to-noise ratio (PSNR), SSIM [95], LPIPS [96] and cleaned FID [97]. To focus on the reconstructed vehicle itself, we only evaluate the foreground pixels using the predicted segmentation mask [86]. For LiDAR simulation, we place the reconstructed vehicle mesh in its original location and perform ray-casting to generate a simulated point cloud. We compare against held out real LiDAR points. We evaluate what fraction of real LiDAR points do not have a corresponding simulated point (*i.e.*, coverage), the average per-ray $\ell_2$ error, Chamfer and Hausdorff distance. See supp. for additional details.

### 4.2 Reconstruction Quality

**Novel View Synthesis on PandaVehicle and MVMC.** We report NVS results on PandaVehicle in Tab. 1 and on MVMC in Tab. 2, along with reconstruction time ($T$) and rendering speed (*i.e.*,

| Method | MSE ↓ | PSNR ↑ | SSIM ↑ | LPIPS ↓ | T (hour) | FPS |
|---|---|---|---|---|---|---|
| NeRF++ [Zhang et al., 2020] | 0.0138 | 20.86 | 0.611 | 0.300 | 4.70 | 0.05 |
| Instant-NGP [Müller et al., 2022] | 0.0095 | 21.68 | 0.641 | 0.319 | 0.05 | 1.14 |
| NeRS [Zhang et al., 2021] | 0.0176 | 18.49 | 0.562 | 0.265 | 1.37 | 3.23 |
| NVDiffRec [Munkberg et al., 2021] | 0.0114 | 20.46 | 0.593 | 0.396 | 1.07 | 51.2* |
| NeuS [Wang et al., 2021] | 0.0115 | 21.37 | 0.640 | 0.247 | 6.25 | 0.02 |
| SI-ViewWarp [Tulsiani et al., 2018] | 0.0233 | 17.51 | 0.514 | 0.371 | – | 1.67 |
| SAMP [Engelmann et al., 2017] | 0.0144 | 19.52 | 0.628 | 0.283 | 0.09 | 71.4* |
| CADSim (ours) | **0.0087** | **21.72** | **0.674** | **0.220** | 0.13 | 49.6* |

Table 1: Evaluation of novel-view synthesis (left → front-left camera) on PandaSet. We report the average GPU hour $T$ for reconstruction per actor and rendering FPS on single RTX A5000. Underline denotes fast reconstruction (*i.e.*, < 10 minutes) and real-time rendering (FPS > 30).

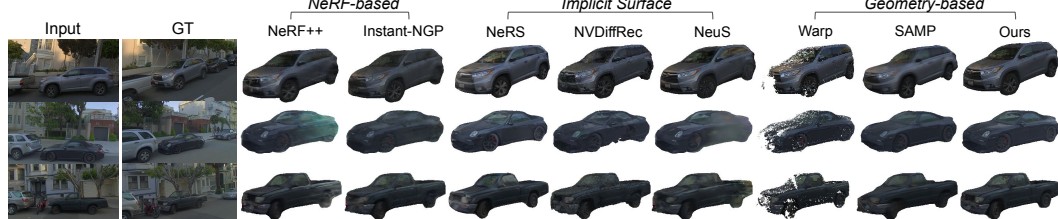

Figure 4: **Qualitative results on PandaVehicle for novel view synthesis**. Compared to existing reconstruction approaches, CADSim produces more robust and realistic results on large extrapolation.

frame per second (FPS)). For PandaVehicle we train each approach using images only from the left camera, and evaluate on front-left and front cameras. For MVMC we evaluate each approach via a cross-validation approach, where we hold out one image each time. CADSim achieves better performance across all image quality metrics, especially LPIPS. LPIPS measures how neural nets perceive images. This suggests that our simulation results are perceived by the ML models to be more similar to real-world images (see Sec 4.3 for more analysis).

With CAD priors, we can reconstruct from much fewer views compared to NeRF-based or implicit-surface based-approaches. Compared to geometry-based methods, our differentiable renderer and material model enable much higher photorealism. Qualitative results on PandaVehicle in Fig. 4 further confirm this. NeRF-based approaches usually lead to visible artifacts with large view extrapolation and sparse input observations. Implicit surface approaches perform better by alleviating shape-radiance ambiguity [43], but still have artifacts due to lower quality

| Method | MSE ↓ | PSNR ↑ | SSIM ↑ | LPIPS ↓ | FIDS ↓ |
|---|---|---|---|---|---|
| NeRS [30] | 0.0254 | 16.5 | 0.720 | 0.172 | 60.9 |
| CADSim | **0.0188** | **17.7** | **0.751** | **0.147** | **48.1** |

Table 2: Evaluation of NVS on MVMC [30].

| Geometry | $L_2$ error ↓ | Hit rate ↑ | Chamfer ↓ | Hausdorff ↓ |
|---|---|---|---|---|
| NeRS [30] | 0.171 | 94.3% | 0.249 | 1.028 |
| NVDiffRec [67] | 0.320 | **98.2%** | 0.439 | 1.708 |
| NeuS [48] | 0.367 | 90.3% | 0.424 | 1.151 |
| SAMP [31] | 0.158 | 94.8% | 0.256 | 1.043 |
| CADSim (ours) | **0.151** | 96.3% | **0.245** | **0.972** |

Table 3: Comparison of LiDAR rendering metrics.

geometries. The other geometry-based approaches have better geometry, but have blurry appearance or missing pixels compared to CADSim. To further demonstrate the value of our improved geometry, we compare the NVS results using our reconstructed mesh against a NeRS mesh with various appearance representations in the supp. material (Tab. A1). Additionally, neural rendering based approaches such as NeuS and NeRF++ are usually slow to train and render. While Instant-NGP [46] reduces training time significantly, it does not support real-time rendering. We report the rendering time for NVDiffRec, SAMP and CADSim (highlighted with *) using the differentiable renderer Nvdiffrast [98]. Faster rendering (>100 FPS) is expected with other modern graphics engines.

**LiDAR simulation on PandaVehicle.** We now evaluate the geometry performance of the reconstructed meshes for LiDAR simulation. As shown in Tab. 3, our approach is better than or comparable to prior art across all metrics, suggesting that our meshes are more complete and more accurate than existing methods. NVDiffRec has the highest hit rate, yet performs badly on the rest of the metrics. This is because their generated meshes are noisy and cover many LiDAR points.

**Ablation study.** To showcase the importance of CAD initalization, we initalize the mesh with the following alternatives: a unit sphere, a rescaled ellipsoid, and geometries from either SAMP [31] or NeRS [30]. As shown in Fig. 5, exploiting CAD priors leads to faster convergence and better performance. Our reconstruction is able to capture fine-grained details such as wheel size and rotation,

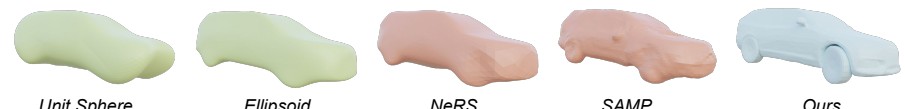

| Unit Sphere | Ellipsoid | NeRS | SAMP | Ours |

Figure 5: CAD priors are crucial for achieving high-fidelity shape in non-convex optimization.

| | Left camera → Front-left camera | | | | Left camera → Front camera | | | |
| | Blending [17] | | Copy-Paste | | Blending [17] | | Copy-Paste | |
| | Det. (IoU) | Segm. (IoU) | Det. (IoU) | Segm. (IoU) | Det. (IoU) | Segm. (IoU) | Det. (IoU) | Segm. (IoU) |
|---|---|---|---|---|---|---|---|---|
| Instant-NGP | 86.77 | 86.86 | 80.57 | 80.81 | 66.37 | 65.80 | 41.65 | 40.13 |
| NeuS | 93.97 | **94.22** | 92.82 | 93.63 | 66.26 | 65.09 | 66.01 | 64.66 |
| SAMP | 90.39 | 89.58 | 90.04 | 89.73 | 82.82 | 79.78 | 83.32 | **81.79** |
| CADSim (ours) | **94.15** | 93.92 | **93.71** | **93.72** | **83.99** | **80.94** | **83.45** | 81.42 |

Table 4: Evaluation of downstream perception tasks (*i.e.*, object detection, instance segmentation) on camera simulation, metric agreement with real data. Our assets lead to smaller domain gaps.

rear-view mirrors, etc. We also evaluate our reflectance model choice [81] in the supp. material. Our physics-based material and lighting model performs the best and generalizes well to novel views.

### 4.3 Realistic and Controllable Simulation

We now showcase applying CADSim for accurate camera simulation to evaluate perception models, and to perform realistic vehicle insertion. CADSim also supports texture transfer naturally, allowing us to easily expand the asset library. Please see additional details and results in the supp. material.

**Downstream evaluation on camera simulation.** To verify if CADSim helps reduce domain gap for downstream perception tasks, we evaluate object detection and instance segmentation algorithms [99, 86] on simulated camera images at novel views. In particular, we consider two setups: rendered and blended [17] and copy-pasted. We compute the instance-level Intersection over Union (IoU) of the predicted bounding box and segmentation mask between simulated and real images. This detection/segmentation agreement metric (the larger the better) indicates how well the simulated data align with the real data with respect to these two perception tasks and model architectures. This metric can provide guidance on using simulation as one tool for evaluation. As shown in Tab. 4, using CADSim assets usually leads to the largest agreement with real images under different settings.

**Realistic animated vehicle insertion.** Given the original scenario, sensor data, and an inserted actor trajectory, we produce consistent multi-sensor simulation with the rendered asset. In Fig. 6, we generate safety-critical scenarios by inserting a new actor turning into our lane (left half) and an actor slowing down in the adjacent lane, preventing the SDV from lane-changing (right half). The LiDAR simulation is conducted on reconstructed meshes enriched with intensity values (see supp. A2). The rendered asset is seamlessly blended into the original scenario and has physically realistic wheel rotation and movement. This enables diverse scenario generation for end-to-end autonomy testing.

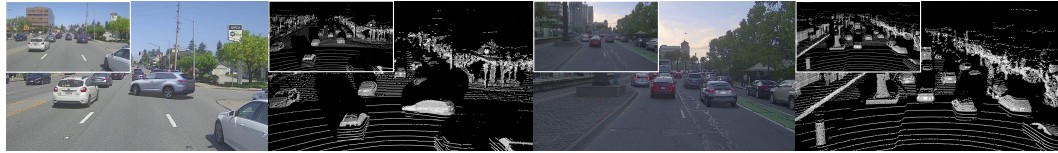

Figure 6: **Multi-sensor simulation on safety-critical scenarios with animated vehicle insertion.** The original camera and LiDAR observations captured in the real world are at the top-left corners.

## 5 Limitations and Conclusion

CADSim's main assumption is that it requires CAD models for the object class of interest. We note that CAD models are readily available for most object classes, and that our approach only requires encoding semantic priors for a single CAD asset, as our energy-model optimization allows for transfer of these priors to other assets of the same class. See supp. for non-vehicle classes. Additionally, our approach relies on either good segmentation masks or LiDAR points to reconstruct accurate shapes.

In this paper, we proposed to leverage in-the-wild camera and LiDAR data to reconstruct objects such as vehicles. Towards this goal, we designed CADSim, which leverages geometry and semantic cues from CAD models with differentiable rendering, to generate meshes with high quality geometry and appearance. These cues also enable our approach to generate *articulated* and *editable* meshes, enabling endless creation of new shapes, textures, and animations for simulation. We demonstrated that integrating our meshes into a camera simulation system can more effectively evaluate perception algorithms than existing reconstruction methods, reducing the simulation-to-real domain gap.

## Acknowledgement

We sincerely thank the anonymous reviewers for their insightful suggestions. We would like to thank Chris Zhang and Yuwen Xiong for their feedback on the early draft and final proofreading. We also thank the Waabi team for their valuable assistance and support.

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
