# OpenReview forum: "CADSim: Robust and Scalable in-the-wild 3D Reconstruction for Controllable Sensor Simulation"
_robot-learning.org/CoRL/2022/Conference — CoRL 2022 Poster_

### Official Review · Reviewer_iXu8 · 2022-07-04

**Originality:** Good
**Technical Quality:** Excellent
**Clarity Of Presentation:** Very Good
**Impact:** 3

**Recommendation:**

Strong Accept: I recommend accepting the paper and will argue for my recommendation even if other reviewers hold a different opinion.

**Summary:**

The paper proposes CADSim, a system that reconstructs vehicle geometry and appearance from multi-view images and optionally multi-view LiDAR pointclouds. The mesh-based model is initialized from a database of CAD models and then fitted to the data (images and pointclouds) by minimizing an energy function in the vertex locations, camera/LiDAR poses, and appearance. The energy function is comprised of data terms capturing the color, mask, and pointcloud consistency as well as multiple regularisation terms. The obtained vehicle models feature articulate wheels and show comparable or better performance than SOTA methods for novel view synthesis and LiDAR rendering. Finally, the obtained models are evaluated wrt. a down-stream perception tas.

**Issues:**

- In Eqn. (5) every point on the current mesh is assumed to have a close point in the point cloud. However, the back half of the car is usually not in the pointcloud. This seems like the "direction" of the asymmetric CD is exactly the opposite of what it should be?
- In 3.3 Inference, Initialization (line 223 ff.) it is not mentioned how the optimization problem over $z$ is solved.
- Instant-NGP should deliver more than 1.14 FPS (Table 1)
- For a simulator I consider real-time rendering to be less crucial (lines 5 and 39).
- There are a few works that could be cited in the related work:
  - Chabot et al. (2017). “Deep manta: A coarse-to-fine many-task network for joint 2d and 3d vehicle analysis from monocular image.” (Also uses a CAD database)
  - Engelmann et al. (2016). “Joint object pose estimation and shape reconstruction in urban street scenes using 3D shape priors.” (Uses a latent code based on CAD models)
  - Wang et al. (2019). “DirectShape: Photometric Alignment of Shape Priors for Visual Vehicle Pose and Shape Estimation.” (similar as above but w/ photometric loss)
  - Koestler et al. (2020). "Learning monocular 3D vehicle detection without 3D bounding box labels." (similar as above but in a DL framework)
  - Loper, M. M. and M. J. Black (2014). “OpenDR: An approximate differentiable renderer.” (early differentiable renderer)
- For the smooth L1 loss one should cite prior work instead of Fast R-CNN: Huber, P. J. (1964). “Robust Estimation of a Location Parameter.”
- E_LiDAR is capitalized inconsistently.

**Quality Of The Limitations Section:**

Limitations are addressed clearly

**Reviewer Expertise:**

3: The reviewer is fairly confident that the evaluation is correct

**Robotics Focus:**

Highly relevant to robotics but no hardware experiments

**Strengths And Weaknesses:**

**Strengths:**

- The paper is well-written, clearly understandable, and of high visual quality. This is especially respectable given that the paper touches many fields, e.g. (neural) rendering, geometric losses, mesh & appearance parametrization.
- While the idea of fitting mesh-based representations to data using CAD models as priors is not very novel in itself, the proposed method is sound and delivers clear advantages wrt. the final performance and practical usability such as fast rendering, editability, and compatibility with current rendering/game engines.
- The paper tackles real-world data and shows high-quality results.
- The supplementary material and the paper's references are extensive.


**Weaknesses**:

- The paper does not promise to publish open-source code. Given the very involved architecture of the system, it is likely that reproduction by a third party would either take considerable time or would fail to achieve similar results. It would be very beneficial for the community if the paper's code could be released.
- Given the intricate nature of the proposed system, it is likely less extendable than, for example, current neural-fields-based systems.
- Given the intricate nature of the proposed system and required prior knowledge, the paper is very information-dense and hence not an easy read. While I do not generally see this as a weakness, because this can be the nature of research, this might limit the number of follow-up works building on the proposed method.



**Summary Of Recommendation:**

Overall, the paper is of very high quality and a very good contribution. The strengths clearly outweigh the weaknesses.

---

### Official Review · Reviewer_gqRW · 2022-07-29

**Originality:** Very Good
**Technical Quality:** Excellent
**Clarity Of Presentation:** Very Good
**Impact:** 3

**Recommendation:**

Strong Accept: I recommend accepting the paper and will argue for my recommendation even if other reviewers hold a different opinion.

**Summary:**

This paper presents a novel method for extracting 3D meshes for a specific category from videos taken in the wild. Instead of relying on a sphere as a starting point for their reconstruction, they use the LIDAR data to find a close matching CAD model of the current category, which is then adapted to be as close as possible to the real object. This reliance on CAD models allows them further to predict each model's texture and lighting mode. Lastly, through the use of these decomposed CAD models, they manage to extract the wheel rotation independent of the car orientation. The evaluation is done by comparing the reconstruction performance with typical object and scene reconstruction methods like NERF++ or NeRS.



**Issues:**

* clarify the statement in 4.3

**Quality Of The Limitations Section:**

Limitations are addressed clearly

**Reviewer Expertise:**

4: The reviewer is confident but not absolutely certain that the evaluation is correct

**Robotics Focus:**

Highly relevant to robotics but no hardware experiments

**Strengths And Weaknesses:**

Strengths:
* The paper is well written and nicely presented
* The supplementary video is outstanding and provides a nice overview with an excellent presentation of the results
* I particular like the decoupling of the model into different parts, allowing for later manipulation of those


Weaknesses:
* While the evaluation compared to other methods is done, the paper claims that these models can be used to train other machine learning methods. However, this claim was not evaluated in the paper. But I can see how the manifold of different evaluations already was enough for one publication.
* I am confused about 4.3, where the downstream evaluation on camera simulated data is done. Here, the extracted 3D models are rendered back into images and tested on different DL methods. Their performance is then compared between the real images and the simulated ones. They then state: "how well we can use the simulation to test existing perception systems". The problem with this is that this claim seems to state that we could use the simulated data to evaluate any future system and avoid using real data, which is of course, the dream. However, I do not believe this claim can be made based on the presented results, here the test is limited to two scenarios, which only shows that the used networks do well in this task, even for simulated data. Not that any other method would also do equally well on this simulated data. Could you please rephrase that statement.



**Summary Of Recommendation:**

Even though I am not happy with a few things in the paper. I overall like the presentation and the concept of the paper very much. I think it would be a valuable addition to CORL. I would still love to hear an answer to my two single weaknesses, but I can understand that a paper can not include every test possible under the sun.

---

### Official Review · Reviewer_NPXi · 2022-07-31

**Originality:** Good
**Technical Quality:** Good
**Clarity Of Presentation:** Very Good
**Impact:** 4

**Recommendation:**

Weak Accept: I recommend accepting the paper, but will not argue for my recommendation if the majority of other reviewers have a different opinion.

**Summary:**

This paper proposes a 3D reconstruction pipeline motivated by the need to simulate different sensors in data-driven simulated environments. The method employs CAD models as priors in improving 3D reconstruction quality. The result is shown generating more complete models that are compatible with the gamut of visual processing augmentations including re-animation and texture transfer.

**Issues:**

-

**Quality Of The Limitations Section:**

Limitations are addressed clearly

**Reviewer Expertise:**

4: The reviewer is confident but not absolutely certain that the evaluation is correct

**Robotics Focus:**

Sufficient demonstration on hardware

**Strengths And Weaknesses:**

The paper tackles an important problem that should be of interest to a substantial portion of the robotics community.

The results are visually impressive, including recovery of complete 3D geometry from incomplete scans and simulation of camera and LiDAR in complex scenes.

The task this paper has set out to accomplish is fundamentally that of reconstructing 3D scenes from sensor data. This is a very well explored space. Note that all methods capable of building a model and rendering novel views are carrying out sensor simulation (of cameras) and are generally trivially extended to simulate LiDAR.

As divulged briefly in the literature review (lines 95-104), extensive work employing 3D CAD model priors has appeared throughout the graphics and vision communities. The revised paper expands its treatment of related work, especially around use of CAD priors, and positions itself relative to existing work in this very active area of research. It also includes in the supplement a consideration of robustness to data noise including, importantly for a paper about sensor simulation, robustness to sensor noise.

The system-level approach here is novel, and introducing techniques from the graphics and computer vision communities into robotics is also of value.

**Summary Of Recommendation:**

The system-level approach here is novel, and introducing techniques from the graphics and computer vision communities into robotics is also of value. I therefore intend to upgrade my review to a weak accept.

---

### Official Review · Reviewer_wSHy · 2022-07-31

**Originality:** Good
**Technical Quality:** Good
**Clarity Of Presentation:** Very Good
**Impact:** 3

**Recommendation:**

Weak Accept: I recommend accepting the paper, but will not argue for my recommendation if the majority of other reviewers have a different opinion.

**Summary:**

This paper presents a new pipeline to provide realistic and controllable sensor (camera and Lidar) simulation to generate in-the-wild data for autonomous driving scenarios.

**Issues:**

1. Table 1 reports training time and rendering time for the system and its comparison methods. Is the computation time of all these methods tested on the same platform as the proposed method or taken from the respective paper/report?

2. In section 3.2, it is explained that the shape is initialized from learned low-dimensional latent code. Does the zero-code initiation always work? From the supplementary video, the pickup truck and the shape generated from the mean code seem quite different. Can the zero code still be optimised for the truck shape? Or is it necessary to have a network to predict a latent code?

3. Generalization: The model representation in 3.1 is only applicable to the vehicle category. It is totally fine as the supplementary document also shows the reconstruction on motorcycles and cones. However, I am not sure if it is correct that a motorcycle only has 3 parts. The reconstruction quality of motorcycles also seems much worse than vehicles. I am curious if the authors have dug into the reasons behind it.

4. It is a bit unclear to me how the pose of the vehicle model is initiliazed. I believe a correct alignment of the CAD model to a detected shape is important for the proposed optimisation pipeline to work. However, there seems to be no analysis on this point.

4. Dynamic scenes: the supplementary material shows 3D reconstruction in static scenes and mixed reality in dynamic scenes.  Does it mean the proposed method can not currently generate 3D reconstruction in dynamic scenes yet? It is unclear in the current version.

Another issue that does not need to be addressed but is worth mentioning:\
* I am not sure if high-fidelity reconstruction is the only metric to care about in the data generation. Sensor noise and distortion are also important when simulating real-world data and testing whether the self-driving vehicles can be robust to those outliers and distortion.

**Quality Of The Limitations Section:**

Additional details required

**Reviewer Expertise:**

4: The reviewer is confident but not absolutely certain that the evaluation is correct

**Robotics Focus:**

Highly relevant to robotics but no hardware experiments

**Strengths And Weaknesses:**

Strengths:
* The paper is clearly written and easy to follow and understand the contributions.
* This paper is a system paper. The method in section 3.2 is not that novel, but the combination works well and shows improvement compared to previous approaches. The semantic information-driven model decomposition in section 3.1 is useful to add variety to the data generation.
* The training time required for the system is not large, and it can achieve real-time rendering speed.
* The supplementary materials are very informative and contain sufficient explanations to help understand this work.


Weakness:
* Several issues need to be addressed. They are raised below.
* One motivation emphasised in the paper is that the exisiting works are sensitive to sensor noise and distortion. However, the experiments did not focus on this point. The input images seem to all have good lighting conditions, and LiDAR motion distortion is not obvious in the test data.
* The apperance term in L216-218 poses a strong assumption on the surrounding lighting condition. It raises the question of whether the proposed simulation pipeline can be generalised to different scenes. For example, light reflection and shading on the vehicles at night would probably violate this term.
* The tests are conducted mainly on the dataset benchmark. It is unclear what the required data inputs are for the system. For example, are sensor pose and object annotations required to generate data in this pipeline? It will be better if some real-world experiments can be conducted on an SDV platform.

**Summary Of Recommendation:**

This paper is a good system paper and presents a realistic sensor simulation tool for in-the-wild data generation. However, I still have a few concerns regarding this work after reading it. It will be better if those concerns can be addressed in the rebuttal.

---

### Meta-Review · Area_Chair_ERxB · 2022-08-10

**Recommendation:** Accept (Poster)
**Confidence:** 4

**Metareview:**

This paper is well written and addresses a problem relevant to the scope of CoRL. The results with a real robot demonstrate the effectiveness of the proposed approach.

Please try to address the reviewers' concerns raised during rethe buttal including a better presentation of the paper's novelty and contributions, model robustness, downstream training using simulation, Extendability of the system, and others raised by reviewers.





**Best Paper Nomination:**

No